# Environmental Sounds Influence the Multisensory Perception of Chocolate Gelati

**DOI:** 10.3390/foods8040124

**Published:** 2019-04-15

**Authors:** Yi Hsuan Tiffany Lin, Nazimah Hamid, Daniel Shepherd, Kevin Kantono, Charles Spence

**Affiliations:** 1Department of Food Science, Auckland University of Technology, Private Bag 92006, Auckland 1142, New Zealand; tlin@aut.ac.nz (Y.H.T.L.); kkantono@aut.ac.nz (K.K.); 2Department of Psychology, Auckland University of Technology, Private Bag 92006, Auckland 1142, New Zealand; daniel.shepherd@aut.ac.nz; 3Crossmodal Research Laboratory, Department of Experimental Psychology, Anna Watts Building, University of Oxford, Oxford OX2 6GG, UK; charles.spence@psy.ox.ac.uk

**Keywords:** TCATA, crossmodal, core affect, psychoacoustics, ice cream

## Abstract

Recently, it has been shown that various auditory stimuli modulate flavour perception. The present study attempts to understand the effects of environmental sounds (park, food court, fast food restaurant, cafe, and bar sounds) on the perception of chocolate gelato (specifically, sweet, bitter, milky, creamy, cocoa, roasted, and vanilla notes) using the Temporal Check-All-That-Apply (TCATA) method. Additionally, affective ratings of the auditory stimuli were obtained using the Self-Assessment Manikin (SAM) in terms of their valence, arousal, and dominance. In total, 58 panellists rated the sounds and chocolate gelato in a sensory laboratory. The results revealed that bitterness, roasted, and cocoa notes were more evident when the bar, fast food, and food court sounds were played. Meanwhile, sweetness was cited more in the early mastication period when listening to park and café sounds. The park sound was significantly higher in valence, while the bar sound was significantly higher in arousal. Dominance was significantly higher for the fast food restaurant, food court, and bar sound conditions. Intriguingly, the valence evoked by the pleasant park sound was positively correlated with the sweetness of the gelato. Meanwhile, the arousal associated with bar sounds was positively correlated with bitterness, roasted, and cocoa attributes. Taken together, these results clearly demonstrate that people’s perception of the flavour of gelato varied with the different real-world sounds used in this study.

## 1. Introduction

Sounds influence consumer consumption behaviour as well as consumers’ flavour and hedonic perception of food. Over the last decade, researchers have been exploring crossmodal correspondences between sounds and the taste, aroma, flavour and/or texture of foods. Studies investigating the effects of audition on other senses have examined the influence of auditory cues on odours such as roasted coffee [1]; potato-chips and coffee [2] and vanilla [3]; tastes such as bitter and sweet [3,4,5,6,7,8,9,10,11,12,13], and; textures such as crunchiness [14], crispness [15], and carbonation [16]. These findings have important implications as sound has the potential to be modulated to enhance the consumer’s eating experience by manipulating the emotional congruency between sound and food perception. However, these studies have only examined the sound-food relationship based on single measures taken at the end of tasting/a consumption episode. Given that the overall perception of food is multisensory and by its very nature dynamic, the influence of auditory cues will likely depend on the integration of multiple sensory attributes that evolve over time.

As an alternative to discrete measures, continuous dynamic measures can be used. Recently, researchers have investigated the dynamic effects of sounds on the perception of food using time intensity (TI), Temporal Dominance of Sensations (TDS) and Temporal Check-All-That-Apply (TCATA) methods. Music varying in liking has been shown to influence the flavour perception of ice cream using TI [17], TDS [18,19] and TCATA [20]. Kantono et al. [18,19] also found that emotions evoked by listening to music varying in valence were found to influence flavour perception. The effects of sounds on temporal changes in flavour however is limited. To date, only TI has been used to investigate the effects of sweet-sour and sour-sweet sounds [21,22] as well eating environmental sounds [23]. These sounds have been shown to influence the perception of taste and pleasantness of food, respectively. The TCATA method affords concurrent selection of multiple relevant attributes, whereas the TDS approach captures a sequence of dominant attributes instead [24,25]. Hence the TCATA method arguably describes the dynamic sensory profiles of products in greater detail than TDS as it permits the documentation of several sensory attributes concurrently [26]. In addition, a recent study [27] showed that TCATA showed higher discrimination and panel agreement levels compared to TDS. 

According to [28], emotions can be described using three specific dimensions: valence (i.e., the pleasantness of stimulus), arousal (i.e., the intensity of the emotions elicited by a stimulus), and dominance (i.e., the degree of attentional control exerted by a stimulus). The arousal and valence states of an individual can be influenced by listening to classical music [29], soundscapes [30,31,32] the eating environment sounds [33,34,35,36]. Most of the research in this area has, however, focused only on the core affect of valence and arousal, often neglecting the dominance dimension. Kantono et al. [17] attempted to map the affective responses of different eating environment sounds, and investigate their influence on the pleasantness of gelato over time using a time-intensity approach. They reported that a café soundscape, which evoked the highest pleasantness ratings, had the lowest arousal and dominance as compared to fast food restaurant and bar sounds. The café sound also had the highest temporal pleasantness rating compared to the low valence, highly arousing and dominant fast food restaurant and bar sounds.

It has been established in our previous studies [17,18,19,20] that changes in the sensory properties of gelato do indeed vary over time when consumed while listening to music that differed in terms of its valence. This implies that the positive and negative valence of auditory stimuli may drive crossmodal associations. Hence, in the present study, the effects of various environmental sounds taken from representative locations where food is frequently consumed (park, food court, fast food restaurant, cafe, and bar sounds) on the temporal perception of the flavour of chocolate gelato using TCATA in a sensory laboratory setting were investigated. In addition, a psychoacoustical analysis of the sounds used in this study will be undertaken alongside the collection of core affect ratings from panellists exposed to these sounds. We hypothesise that each environmental sound will evoke a specific affective state, which will then influence the temporal flavour profile of chocolate gelato.

## 2. Material and Methods

### 2.1. Ethics Statement

Ethics approval by the Auckland University of Technology Ethics Committee (AUTEC 12/79) was obtained for this study. All of the panellists signed informed consent forms prior to the commencement of the study.

### 2.2. Panellists

Fifty-eight trained panellists (21 males, 37 females) between 18 and 41 years of age ($\bar{x}$ = 25.6, σ = 3.7) took part in this study. All 58 panellists carried out evaluations under the control and five sound conditions. A minimum of 40 participants were required for each condition to achieve a statistical power level of 0.90–0.95. This estimate was based on Cohen’s calculation using α = 0.05 and β = 0.2. Panellists were recruited online through an advertisement posted on social networking services (i.e., Facebook and Instagram), and were rewarded with supermarket vouchers for their participation. None of the panellists were smokers, experienced hearing loss, suffered from any eating disorders, or other health problems associated with food. Data collection occurred over a three week period. Both training and evaluation was done in a sensory laboratory at Auckland University of Technology. 

### 2.3. Background Sounds

Background noises were recorded between 01:00 p.m. and 02:00 p.m. on the same day of the week (Monday) in five different settings located in Auckland Central Business District, New Zealand: a café, a fast food restaurant, a bar, a food court, and a park. The Root Mean Square amplitudes of the audio samples were standardized to an internal reference in order to achieve equivalent average sound pressure levels across all audio samples, and later scaled to 70 dB of sound pressure level (SPL), using a Brüel and Kjær sound meter (Brüel & Kjær, Nærum, Denmark).The audio samples were played through a standard PC soundcard driving a Sennheiser headset (Series HD 518, Sennheiser Electronics GmbH & Co. KG, Wiedermark, Germany). Sound presentation was randomized, and counterbalanced across panellists [37]. The same gelato was consumed in all five sound conditions and in the silent control condition.

### 2.4. Sample Preparation and Presentation

Chocolate flavoured gelato was chosen in this study because chocolate has been reported to be an emotional food that is influenced by background music in terms of overall impression [38]. The chocolate gelato samples were made with cream (40%), milk (30%), sugar (15%), and cocoa powder (15%) using an ice cream maker (Cuisinart ICE-100 Compressor Ice Cream and Gelato Maker, Cuisinart, Stamford, CT, USA). Prepared samples were placed in polystyrene cups and then frozen in a commercial-grade freezer (Fisher and Paykel, East Tamaki, New Zealand) at −18 °C for at least 24 hours prior to testing in order to ensure sample consistency. Prior to serving, all samples were tempered for five minutes at room temperature. The serving temperature (−12 ± 2 °C) was strictly monitored to maintain consistency [39]. A scoop of frozen gelato (5.0 ± 0.8 g) was then placed individually into a sealed white plastic container (45 mm diameter) coded with a three-digit random number before being served to the panellists. Each sample was tempered 5 min before being served to panellists for tasting. 

The panellists were given a 30 s break in-between samples and instructed to drink water to cleanse their palate. This was reinforced by having a screen where each panellist had a forced break of 30 s and in which panellists were required to drink water during this time in silence. The time interval between samples was determined by several pilot trials to ensure no residual carryover of flavour occurred before the next sample was tasted. The chocolate gelato used in this study was also specially formulated so as to melt slowly in the mouth with no strong bitter after taste sensations persisting before the next sample was tasted. 

### 2.5. Panel Training

Panel training totalling 10 hours was carried out over three sessions. A commercially available chocolate ice cream was used for training purposes. Panellists were informed that they would be listening to various sounds while consuming chocolate ice cream. Panellists first familiarised themselves with temporal measurements of sensations using TCATA, and were reminded to attend to the multiple attributes that were perceived in the product and to continuously select the attributes that were present and deselect those that were absent. Panellists were also asked to familiarise themselves with both the sensory and affective attribute definitions (Table 1 and Table 2, respectively). Prior to evaluation, the panellists were emailed a demonstration video created in-house that described the TCATA procedure. This video was shown to the panellists once again when they arrived at the laboratory. Before participating in the real test sessions, the panellists received a dummy gelato sample and performed TCATA ratings on it in a warm-up session.

### 2.6. Temporal Check-All-That-Apply (TCATA)

Temporal Check-All-That-Apply (TCATA), developed by [24], was used to document flavour changes in chocolate gelato over a specified time while listening to the different sounds. In this method, multiple attributes can be selected simultaneously, thus permitting the description of sensations that arise either sequentially or concurrently [41]. The TCATA procedure in this study adapted the protocol reported by [42,43,44]. The modification involved intensity scales being replaced with buttons that corresponded to the sensory attributes used in this study. The TCATA data was coded as binary values over time (0 for unchecked attributes and 1 for checked attributes). The panellists continually updated the attributes of the sample over time by checking attributes at times whenever applicable, and unchecking attributes, whenever not applicable. The seven sensory attributes that best represent the flavour of chocolate gelati used in this study can be found in Table 1, and were based on the studies by [18,20]. In this study, the attributes were selected based on a focus group who identified the most important attributes in chocolate gelati sample and how they changed over time.

### 2.7. Affective Responses to Background Sounds

An affective state is considered an instinctual reaction to stimuli, and is typically modelled using three dimensions: valence, arousal, and dominance. According to [40], arousal represents the level of arousal or excitement elicited by a stimulus, and is measured either by self-report or by electrophysiological recordings of the sympathetic nervous system. Valence represents the perceived pleasantness, while the dominance dimension represents the assertiveness of the perceived stimulus and its capability to capture attention. Panellist affective states were measured using the Self-Assessment Manikin (SAM) as described by Bradley and Lang [45]. A nine-point categorical scale was used to measure valence (unpleasant–pleasant), arousal (calm–excited), and dominance (least dominant–most dominant).

### 2.8. Experimental Procedure

Once panellists completed their training they were invited to attend a separate session for the actual evaluation of gelato samples using the TCATA method. The panellists were informed at the start of each trial to taste the gelato sample and then select the appropriate sensations while listening to the sounds. Panellists selected attributes by choosing the appropriate buttons displayed on a screen. Each sound was automatically played as soon as the participants first clicked the TCATA button provided on screen. The following on-screen instructions were then displayed: “Please place sample in mouth for the first 5 seconds” and “Please swallow the sample at the fifth second of the trial”. This was done in order to try and reduce individual variability in people’s eating behaviours, and thus to ensure that each panellist experienced the gelato in as similar a manner as possible. When a new attribute was perceived, the corresponding button was selected. Panellists were permitted to select the same attribute repeatedly, or not select an attribute at all, concurrently, over a period of 45 seconds. The FIZZ Acquisition software (version 2.46b, Biosystemes, Saint-Ouen-l’Aumône, France) recorded the ratings at a sampling rate of 5 Hz (i.e., approximately five data points per second). After temporal evaluation of all sound-food pairs, panellists were asked to report their affective responses (i.e., valence, arousal, and dominance) to the sounds using a 9-point categorical SAM scale. A repeated-measures design was used, with all participants exposed to all five sound conditions as well as the silent control condition when tasting the gelato. A summary of the experimental procedure can be found on Figure 1.

### 2.9. Data Analysis

All univariate and multivariate analysis in this study was carried out using XLSTAT (Addinsoft, Long Island City, NY, USA)

#### 2.9.1. Temporal Check-All-That-Apply (TCATA) Curves

Temporal Check-All-That-Apply (TCATA) curves were generated using the FIZZ software (version 2.46b). Temporal curves depict the proportion of panellists who cite the attribute at a given time. The higher the citation rate for the attribute, the higher the frequency of citation agreement amongst the panellists. Spline-based smoothing was applied on each curve [46] in order to aid visualisation of the TCATA curves.

Analysis of TCATA curves was carried out as described by [24,47]. Reference lines were calculated based on the significant proportion of the curves that were selected at a level that was significantly greater than chance. Reference lines (bolded and highlighted on the curve) were calculated using the two-sided Fisher–Irwin test [48,49]. The Fisher–Irwin test was carried out in order to investigate the homogeneity of citation proportions of the TCATA curves. If the test showed significance, then the citation was considered not to be selected by chance and displayed as reference lines.

The TCATA time period in this study was presented as standardized time (ST), as this provided a better understanding of perception and greater consensus across the whole panel. Each panellist’s time data was standardised to a score between 0 and 100; 0 representing when they clicked the line scale to start and 100 when recording stopped automatically.

#### 2.9.2. Correspondence Analysis

Correspondence Analysis (CA) was applied to the TCATA data to visualise the sum durations of selected sensory attributes. The sum duration of attributes was obtained by summing up the total CATA counts of each attribute for each product for all panellists as a function of time. A CA enables the projection of sensory attributes onto a simplified oral trajectory and a visual map [24]. In addition, chi-square tests of independence between rows (i.e., attributes) and columns (i.e., sounds) were determined to investigate if sensory perception was linked to the different sound conditions. 

#### 2.9.3. Valence, Arousal, and Dominance (VAD) Measures of Background Sounds

A one-way ANOVA was performed on the valence, arousal, and dominance measures as a function of the sounds. Post hoc Tukey’s honestly significant difference (HSD) was applied if significance was observed (*p* < 0.05).

#### 2.9.4. Psychoacoustic Analysis of the Sounds

In the psychoacoustic analysis, parameters other than sound pressure level were determined to describe a sound. These parameters included tonality, fluctuation strength, roughness, and sharpness, all of which co-vary with human responses to sound. According to [50], tonality provides a measure of the relative content of pure tones in a sound, with noise being an example of a sound low in tonality. Fluctuation strength provides a measure of amplitude modulation; that is, cyclic variations in amplitude, and roughness is a measure of modulation with lower frequencies (15–300 Hz). Sharpness provides a measure of the relative content of high frequencies in a signal. In the current study, these psychoacoustical parameters were calculated using the National Instruments LabVIEW 2013 software (National Instruments, Austin, TX, USA).

#### 2.9.5. Multiple Factor Analysis

Multiple Factor Analysis (MFA) enables the simultaneous analysis of datasets of variables to study the relationship between the observations and variables [51]. In this study, MFA was applied to the TCATA sensory duration measures, as well as affective and psychoacoustics measures obtained from this study. This allowed the relationship between the sensory responses to affective measurements and psychoacoustics qualities of the sound to be explored.

## 3. Results

### 3.1. TCATA Curves

Figure 2 depicts the overall TCATA curves for chocolate gelato consumed while exposed to five different environmental sounds and a silent condition. Only those attributes that reached significance (represented as highlighted lines in the TCATA curves) will be discussed. Sweetness was most cited in the café sound condition between 6 and 20% ST with a decreasing citation rate from 78 to 34%. Sweetness was next most cited in the park sound condition, decreasing between 7 and 10% ST from 69 to 56% citation rate, increasing between 15 and 17% ST (40–45% citation rate), decreasing from 17–30% ST (45–18% citation rate). From 37–0% ST, the citation rates of sweetness remained low between 22 and 33%.

Bitterness was the most cited in the bar condition and increased between 7 and 10% ST, reaching a maximum citation rate of 48%. In the food court condition, bitterness increased from 0 to 8% ST, reaching a maximum citation rate of 34%, and from 63–100% ST. Here, the citation rates hovered between 22 and 40% ST. Finally, bitterness was next most cited in the fast food restaurant sound condition, increasing significantly between 14 and 27% ST (34–45% citation rate), and decreasing from 27 to 35% ST (45–21% citation rate).

Milkiness was the most cited in the fast-food restaurant and bar conditions, with similar citation rates between 50–56% ST. Milkiness was most cited in the bar condition, increasing between 11 and 15% ST (41–56% citation rate), and decreasing from 15 to 18% ST (56–44% citation rate) and 35–53% ST (51–34% citation rate). In the fast-food restaurant condition, milkiness increased significantly between 32 and 35% ST (49–53% citation rate), and decreased from 35–38% ST (53–45% citation rate). From 72 to 100% ST, the citation rates remained low between 35–50%.

Creaminess was cited significantly in the café (17–36% ST) and park (56–88% ST) soundconditions, but at low citation rates of between 20 and 35% and 11–32% respectively. Irrespective of sound condition, vanilla was not frequently cited while consuming the gelato.

Cocoaness was cited the most in the food court condition, increasing between 0 and 18% ST (42–61% citation rate), decreasing from 18 to 44% ST (61–40% citation rate), increasing from 47 to 51% ST (40–58% citation rate), and finally decreasing from 51 to 63% ST (58–35% citation rate). Cocoaness was the second most cited attribute in the silent condition, increasing between 17 and 35% ST (41–57% citation rate), and decreasing from 35 to 47% ST (57–40% CR) and 50–60% ST (32–25% citation rate). This was followed by the fast food restaurant sound condition, where cocoaness increased between 39 and 41% ST (41–48% citation rate), decreasing from 41 to 47% ST (48–39% citation rate), and increasing from 80–100% ST (30–42% citation rate). In the park condition, cocoaness increased between 52 and 55% ST (44–50% citation rate), and decreased from 55 to 63% ST (50–33% citation rate).

Roasted was most cited in the café condition, increasing from 40 to 60% ST (28–57% citation rate), decreasing from 60 to 75% ST (57–31% citation rate); increasing from 78 to 85% ST (39–49% citation rate), and decreasing again from 85 to 91% ST (49–40% citation rate). Roasted was next most cited in the silent condition, increasing between 89 and 97% ST from 36–46% citation rate. Finally, in the bar condition, roasted increased between 48 and 54% ST (31–35% citation rate), and 71–90% ST (34–40% citation rate).

### 3.2. Correspondence Analysis (CA)

To further summarise the TCATA results, CA was carried out on the durations for which the attribute was selected [41,52]. The results shown in Figure 3 highlighted significant differences in terms of the sensory attributes of hte gelato in each sound condition (χ^2^_(30)_ = 109.34; *p* < 0.05). Dimension 1 explained 91.48% of the variability and separated the pleasant park and café sounds that had high negative loadings from the unpleasant food court, fast food, and bar sound conditions that had positive loadings. Roasted, cocoa, and bitter attributes were correlated with bar, fast food, and food court sounds, while sweet and creamy attributes were correlated with park and café sounds. Dimension 2, explaining 7.26% of the variance, further separated all five environmental sounds conditions that had positive loadings from the samples consumed in the silent condition.

Factors 1 (F1) and 2 (F2) illustrated across Figure 4 depict a temporal pattern of attribute citations for gelato consumed under each of the five sound and the silent condition. F1 was associated with the sweet attribute that had high positive scores along this factor and was elicited early in the trial. Roasted was elicited later in the trial and had high negative scores along this factor. All of the gelato samples consumed under the different sound conditions followed a similar flavour evolution in terms of sweet at the start of mastication and ending with roasted for only some samples. F2 mainly explained the variance in the milky attribute that had high negative scores along this factor. Interestingly, bitter, creamy, and cocoa attributes were not explained by F1 and F2. Hence, Factor 3 (F3) was taken into account and plots of F2 against F3 are depicted in Figure 5a–f.

The trajectory plots for the different sound conditions will be discussed separately. In the silent condition, sweet was more cited in the range of 0–7% ST, followed by milky that was cited in mid-trial, and roasted cited at the end of trial. F3 further showed bitter being cited early in the trial (9–22% ST) and cocoa cited after between 24 and 47% ST. In the park sound condition, sweet was cited early, and then milky from 30–35% ST with cocoa at the end of evaluation. In the café sound condition, sweetness was more cited early in the trial, then milky from 35 to 40% ST, and finally roasted was more cited from 40% ST until the end of evaluation. F3 also explained short citations of bitter (12–17%) and creamy (20–21%). In the fast food sound condition, sweet was cited for a short time early on, then milky (29–35% ST), and then roasted was cited from 60 to 80% ST. Along F3, bitter was cited between 16 and 19% ST, and the cocoa at mid-evaluation (52–63% ST). In the food court sound condition, sweet was cited shortly after consumption and then milky (26–29% ST). Interestingly, F3 explained bitterness early on as well (7–12% ST), and from 83% ST until the end of trial. In the bar sound condition, sweet was shortly cited at early in the trial, and roasted cited after mid evaluation from 70 to 90% ST. F3 only explained a short citation of milky at around 35% ST.

### 3.3. Affective Responses

As seen in Figure 6, differences were observed in terms of affective dimensions associated with environmental sounds. Specifically, significant differences in terms of valence (F_(5, 719)_ = 446.81, *p* < 0.001), arousal (F_(5, 719)_ = 4813.76, *p* < 0.001), and dominance (F_(5, 719)_ = 309.28, *p* < 0.001), were noted. The bar sound was rated as being the most arousing and least valent. The park and café sounds, on the other hand, were significantly less arousing, but highly valenced. Dominance was highest for fast food restaurant, food court, and bar sounds. Affective responses evoked by the silent condition were significantly lower in terms of valence, arousal, and dominance.

### 3.4. Psychoacoustic Characteristics

Psychoacoustic parameters of the sound in terms of sharpness, roughness, and fluctuation strength were also analysed. The changes in psychoacoustical parameters of the different sounds were consistent with the valence results obtained (see Table 3). Bar sound, which was the least valent had the highest values for sharpness, roughness, and fluctuation strength. On the other hand, the park sound which was the most valent had the lowest values for all three psychoacoustical qualities.

### 3.5. Multiple Factor Analysis (MFA)

In this study, MFA was further carried out to explore the relationship between the datasets for sensory, affective and psychoacoustic results obtained. The MFA model further supported our TCATA results (see Section 3.2 and Section 3.3). The park and café sounds were associated with sweetness and creaminess, while fast food, food court, and bar sounds were associated with cocoa, roasted, and bitterness.

The MFA model (Figure 7) also revealed the relationships between the sounds and silent conditions in terms of affective and psychoacoustics measures. The bitter, roasted, cocoa attributes that were evoked while listening to fast food, food court, and bar sounds were associated with high arousal ratings, and psychoacoustical parameters of sharpness, roughness, and fluctuation strength. On the other hand, the sweet and creamy sensations evoked by the park and café sounds were mainly associated with high valence.

## 4. Discussion

### 4.1. Environmental Sounds Influenced Affective States

The three affective dimensions (i.e., valence, arousal, and dominance) varied significantly with different sound conditions. The park sound was significantly higher in valence and the least arousing of the environmental sounds. Similarly, Guillén and López Barrio [53] reported the urbanised nature sounds (i.e., a park) to be the most positively valenced as compared to traffic noise, as well as social and commercial sounds. The latter researchers reported that “nature” category sounds consisting of wind, water, natural elements, countryside, rain, and parks were deemed most pleasant. Seo and Hummel [2], meanwhile, reported that pleasant sounds can evoke positive attitudes and feelings of comfort in panellists, which, in turn, may influence their judgment of an accompanying stimulus (in this case gelato) as more positive or pleasant. [2] demonstrated that when participants listened to pleasant sounds (e.g., a baby laughing or jazz music), the pleasantness ratings of odours increased for both pleasant and unpleasant odours. Meanwhile, listening to unpleasant sounds (a baby crying or people screaming) sometimes decreased the pleasantness of both odours.

Bar sound was rated significantly greater in terms of arousal followed by food court and fast food restaurant sounds. Highly arousing sounds are generally categorised as stressors, eliciting unpleasant feelings (i.e., annoyance) and physiological stress reactions, especially at high sound pressure levels [30]. Kluger and Rafaeli [34] examined the relationship between pleasure and arousal in different food service settings. They reported that arousal was highest in fast food restaurants (McDonalds and Burger King) as compared to hotels and medical centres.

In our study, dominance ratings were highest in the bar, food court, and fast food restaurant soundconditions. This concurs with our previous results [23] in which bar sounds were found to dominate more than fast food restaurant and café sounds. Furthermore, Medvedev et al. [31] found that three unnatural sounds (construction, motor bike, and airplane) were rated lower on pleasantness and higher on eventfulness and dominance as compared to the two natural sounds (ocean and birdsong) and a piece of music. Most studies, however, have only focused on the measure of valence and arousal of sounds in environmental settings [29,31,32,34,36]. This reflects the importance of taking into account the measure of dominance when analysing the affective states of environmental sounds as dominance was shown to vary with the eating environment sounds used in this study. It is important to note that the sounds used in this study were equalized to 70 dB SPL to provide a comfortable noise level for participants as sounds above 80dB can result in discomfort or hearing impairment [54] Hence, hearing intense sounds may be detrimental to participants, especially when wearing headphones. In addition, a study by Flamme et al., [55] identified the distribution of typical noise levels present in daily life and reported that the acceptable average sound level range was between 70–76 dB. 

### 4.2. TCATA Profiles of Chocolate Gelati in Different Sound Conditions

#### 4.2.1. Sweetness

Sweetness was the first cited attribute in the early mastication period for the control (i.e., silence) and all environmental sound conditions. Similarly, sweetness was reported as the first cited attribute during the assessment of sweetened coffee samples [56], cocoa chocolate [43], and chocolate dairy desserts [57]. Kantono, et al. [18] further showed that sweetness was always the first dominant attribute when consuming chocolate gelati while listening to music. Sweetness has been reported to be an important factor as far as the discrimination of chocolates is concerned [58]. This probably explains why sweetness was always perceived first.

Sweetness was significantly cited in the café and park sound conditions between 7–10% ST and 6–20% ST respectively, with a higher citation rate in the café sound condition. Kantono et al. [23] reported that listening to liked music under laboratory conditions evoked positive emotions that were associated with the perception of sweetness. They further demonstrated that sweetness was cited significantly more often in a real-world eating environment (gelato shop) than in the laboratory environment, while listening to liked music [20]. In a related study, Carvalho et al. [5] demonstrated that listening to a pleasant ‘smooth’ soundtrack increased the perceived sweetness of chocolate compared to a ‘rough’ soundtrack.

#### 4.2.2. Bitterness

Bitterness was highly cited under the fast food restaurant and bar sound conditions, especially in the early stages of mastication. On the other hand, with the food court sound condition, bitterness was highly cited in the later stages of mastication. In this study, the fast food, food court, and bar sounds were rated low in valence and high in arousal and, furthermore, and were associated with bitterness. Similarly, Kantono et al. [17] demonstrated that listening to disliked music evoked negative emotions resulting in higher bitterness ratings in chocolate gelati. Additionally, Platte et al. [11] also reported an increased rating of bitterness after panellists viewed a negative mood video clip (i.e., sad films). These findings suggest that the influence of sounds on the perception of bitterness is mediated by emotions.

#### 4.2.3. Milkiness

Milkiness was the most significantly cited throughout consumption under the fast-food restaurant sound condition. However, we are aware of no previous research reporting on the perceptual changes of milkiness under different environmental sound condition. However, Petit and Sieffermann [35] reported that the frequency of the milk taste term being used to describe milk-based iced coffee increased in a meeting room environment as compared to a laboratory, a ’hot’ laboratory, and a cafeteria. In addition, Kantono et al. [20] also showed milkiness citation rates to be the highest in the natural eating environment (gelato shop) under a neutral music condition as compared to the laboratory environment. Hence, milkiness is an important attribute when evaluating dairy food assessed under different environmental sound conditions.

#### 4.2.4. Cocoaness

Cocoaness was significantly cited over a longer duration, mainly in the food court sound condition compared to other sound conditions. However, no research has reported on how the perception of cocoa changes under different real-world environmental sounds conditions. At the psychoacoustic level of description, the food court sound in our study had a high sharpness value (see Table 3). Kantono et al. [23] reported a high sharpness value for a bar sound as compared to fast-food and café sounds. However, the authors only evaluated the pleasantness of gelati over time with the different sounds, and not any changes in sensory attributes *per se*. Gelati in their study was in fact significantly lower in maximum pleasantness under the bar sound condition compared to café and fast food conditions.

#### 4.2.5. Roasted

Roasted was most significantly cited from 40% ST until the end of mastication only in the café sound condition. Of note, background sounds of espresso machines were present in the café sound condition in our study. The sounds of the espresso machines might have evoked a contextual effect. Higher-order cognitive processes might thus have heightened roasted perception. Sester et al. [36] demonstrated that alcoholic drinks were selected according to perceptual, semantic, or cognitive associations between the two bar-like environments with either wood or blue furniture. Therefore, it is important to take into account specific sounds present in the environment that may evoke a context when investigating the effect of environmental sounds on temporal sensory perception.

### 4.3. Psychoacoustics Parameters Can Predict Core Affect Measures

Hall et al. [33] explored the relationship between perceptual, psychoacoustical, and acoustical properties of urban sounds. These researchers found that psychoacoustical parameters (i.e., sharpness, roughness, and loudness) were significant predictors of valence, with consistently higher beta coefficients compared to fundamental acoustical properties such as frequency. Beta coefficients measure the strength of effects of independent variables to the dependent variables in a proportional manner [59]. The authors concluded that core affect dimensions (i.e., valence and arousal) were valid metrics in assessing sound qualities and suggested that supplementary measurements (such as psychoacoustical parameters) would be able to capture the rich complexity of sound experiences.

In this study, increases in psychoacoustical parameters such as sharpness, roughness, and fluctuation strength decrease the valence rating of sounds. Bar, fast food and food court sounds in this study that had low valence and high arousal ratings showed higher sharpness and roughness values. According to [50], higher values of sharpness were associated with those sounds containing a greater proportion of high frequencies measured. The sensation of sharpness is considered to be inversely related to pleasantness and positively associated with arousal. Roda et al. [29] further found that increased roughness of classical music increased arousal, similar to findings in this study.

### 4.4. Sounds Can Influence Temporal Flavour Perception

This study explored the three dimensions of affect: valence, arousal, and dominance. Other studies have explored the influence of auditory cues such as sounds and music upon flavour perception. Crisinel and Spence [8] mapped the correspondence between basic taste and flavours with musical notes. Soundscapes were further reported to influence the basic taste of cinder toffee [9], beer [4,6], and ice cream [10,23]. The emotional mediation theory [13,23] was proposed to explain these crossmodal effects. 

The findings reported in the present study revealed that listening to fast food, food court, and bar sounds, which were low in valence and high in both arousal and dominance, enhanced the perception of bitter, roasted, and cocoa attributes. On the other hand, listening to park and café sounds, which were high in valence and low in arousal evoked the attributes of sweetness and creaminess. Kantono et al. [23] was the first to explore how subjective dimensions of valence, arousal and dominance explained the crossmodal effect of sounds and food pleasantness. Kantono et al. [18] further examined the relationship between autonomic nervous system (ANS) responses and people’s perception of the flavour of chocolate gelati while listening to music varying in liking. Their results consistently showed that sweetness and creaminess were associated with low arousal responses, while bitterness and cocoaness ratings were associated with high arousal responses. This result also resonates with findings reported by others [60], exploring emotions and flavour perception in terms of the valence and arousal model using the check-all-that-apply (CATA) methodology. They reported that sweetness and creaminess were associated with valence, while cocoaness and bitterness were proportional to arousal.

This study is one of the few studies in the crossmodal sensory science field that measured dominance in addition to valence and arousal. The findings showed significant differences in dominance ratings of the sounds used in this study, and these difference were found to significantly influence gelato flavour. This finding supports the argument made by [28] that emotion does not only fit onto a circumplex model comprising a two dimensional space based on valence and arousal ratings. This is in agreement with Meiselman [61], and thus it is important to consider the measurement of emotions across multiple dimensions. 

## 5. Conclusions

TCATA described the temporal sensory profile of chocolate gelato consumed while listening to sounds varying in affect. Sounds varying in valence resulted in changes of temporal sensory profile of gelato that can be explained in terms of changes in affective dimensions. Both TCATA and CA showed that bitter, cocoa, and roasted attributes were associated with unpleasant and highly arousing bar and fast food restaurant sounds. In contrast, sweetness was associated with high valence park and café sounds. The psychoacoustical properties of the sounds were likewise found to be related to affective dimensions. The MFA results confirmed the negative relationship between valence ratings and both the psychoacoustical qualities of roughness and sharpness. To conclude, our participants’ perception of the flavour of gelato was influenced by the affective measures and psychoacoustical characteristics of sounds. It would be interesting to determine if the effects reported in this study can be extrapolated to more complex food stimuli. Given the important role played by affective measures and psychoacoustical characteristics of sounds, it would be of interest to further investigate the role of emotions evoked by sounds in influencing flavour perception of food. As sound is only one of the sensory components of a multisensory atmosphere that includes smell, temperature and any visual features, it may of interest to look at how food perception changes in different real eating environments.

## Figures and Tables

**Figure 1 foods-08-00124-f001:**
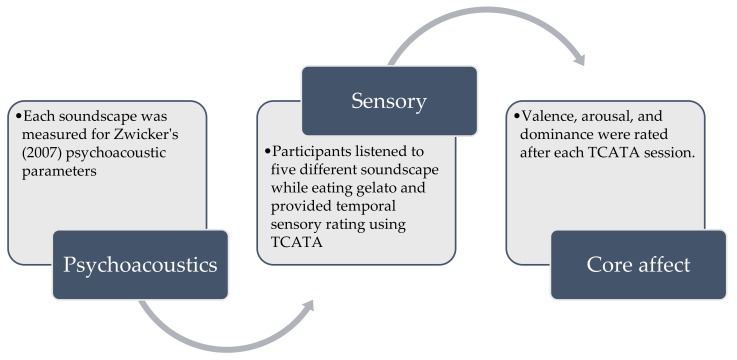
Experimental protocol describing the procedures carried out in this study.

**Figure 2 foods-08-00124-f002:**
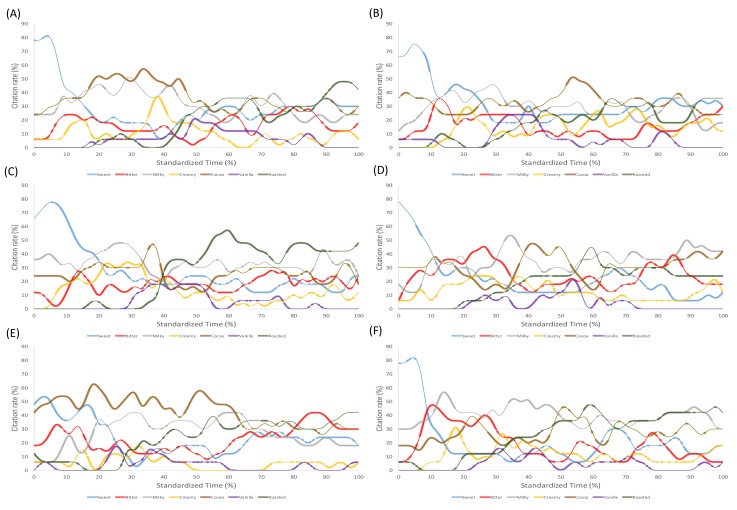
TCATA curves for chocolate gelato consumed under: (**A**) silent, (**B**) park, (**C**) café, (**D**) fast food restaurant, (**E**) food court, and (**F**) bar sound conditions. Reference lines (highlighted) indicate those citation proportions that were statistically significant and not selected by chance.

**Figure 3 foods-08-00124-f003:**
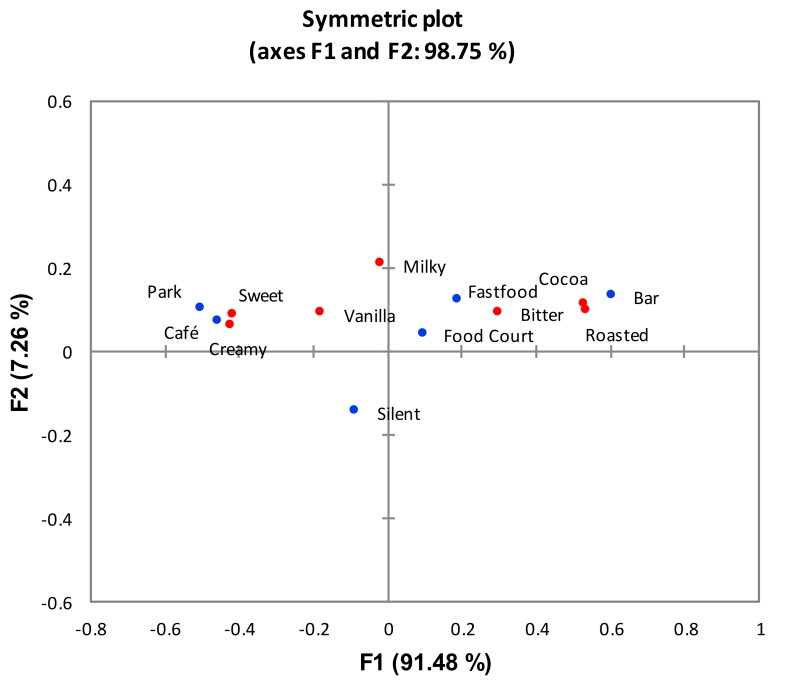
CA factor map (first two components) based on aggregated TCATA data over the whole evaluation duration. Different colours represent the sound conditions (blue) and sensory attributes (red).

**Figure 4 foods-08-00124-f004:**
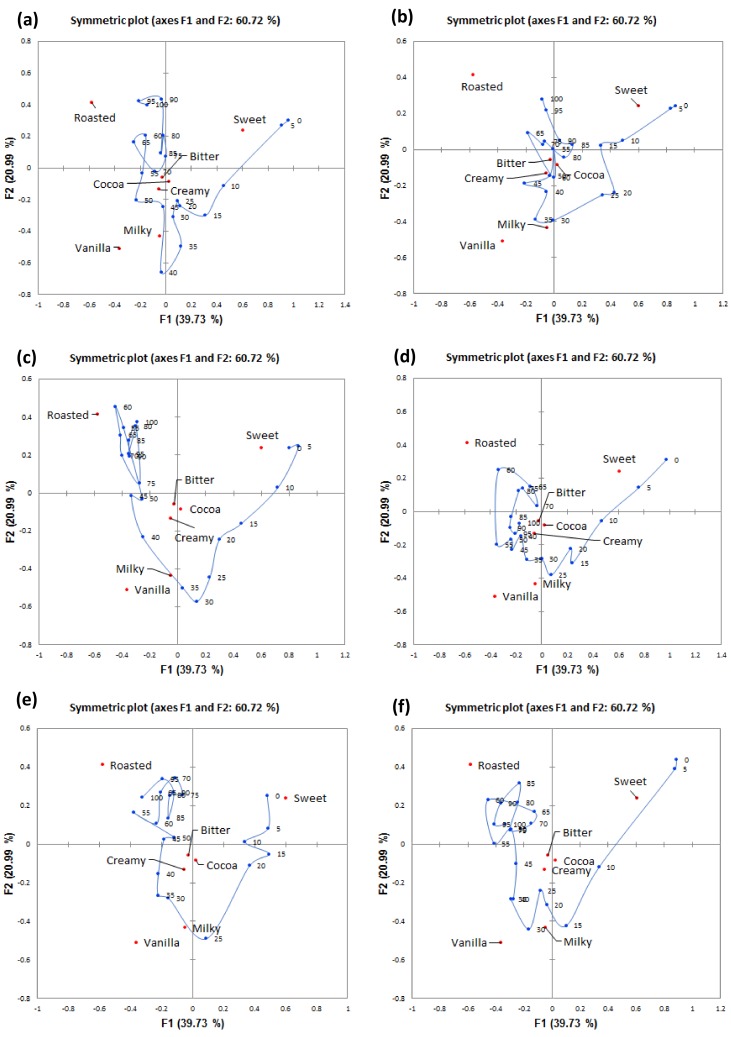
Smoothed oral trajectories from CA factor map (first two factors, F1 and F2) based on TCATA citation proportions of sensory attributes perceived during consumption of gelato under: (**a**) silent, (**b**) park, (**c**) café, (**d**) fast food restaurant, (**e**) food court, and (**f**) bar sound conditions.

**Figure 5 foods-08-00124-f005:**
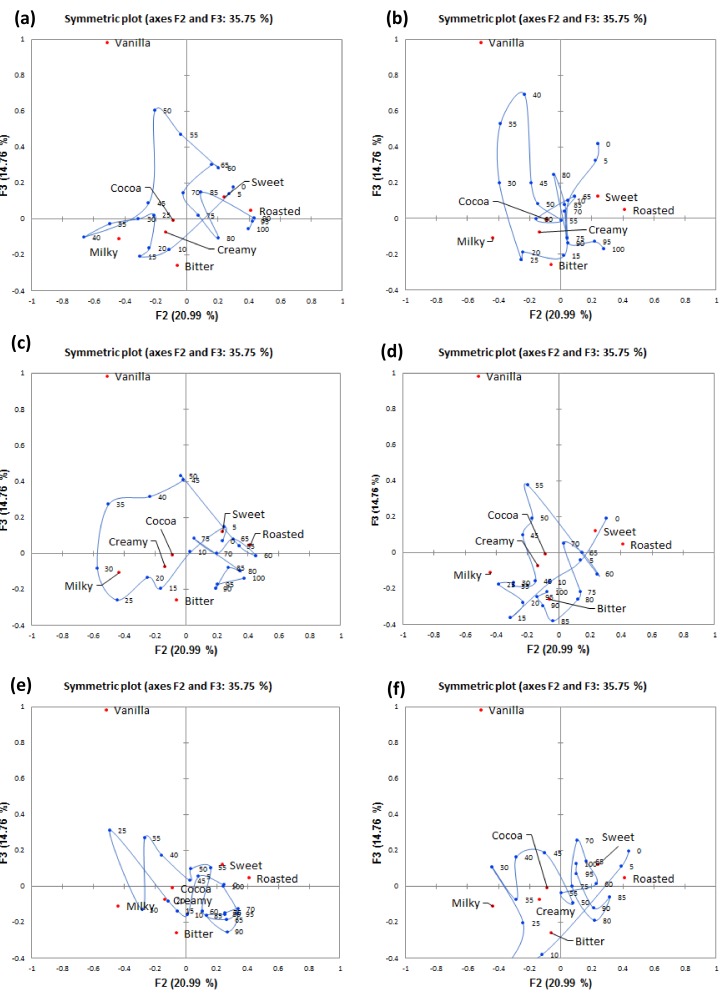
Smoothed oral trajectories from CA factor map (second and third factors, F2 and F3) based on TCATA citation proportions of sensory attributes perceived during consumption of gelato in the: (**a**) silent, (**b**) park, (**c**) café, (**d**) fast-food restaurant, (**e**) food court, and (**f**) bar sound conditions.

**Figure 6 foods-08-00124-f006:**
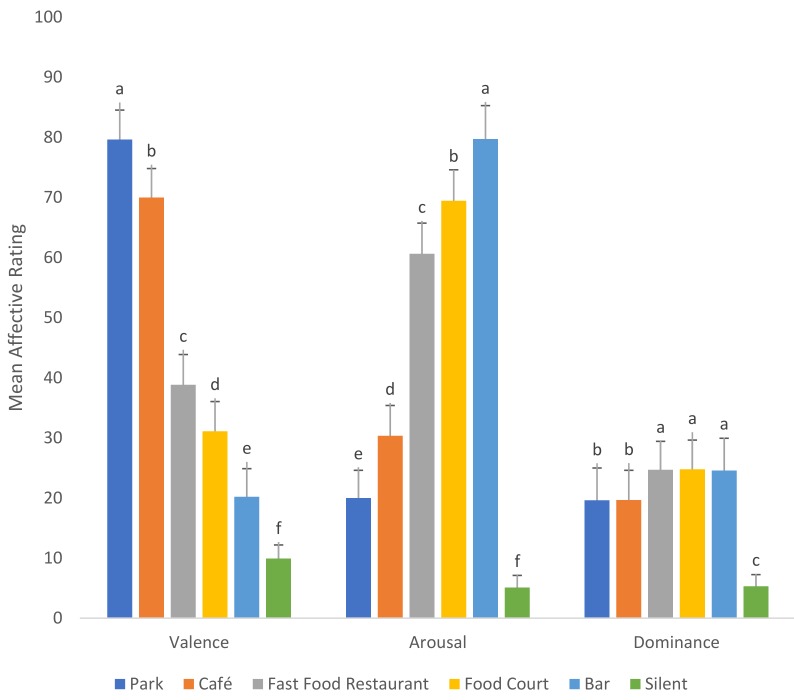
Differences in valence, arousal, and dominance of the various environmental sound conditions.

**Figure 7 foods-08-00124-f007:**
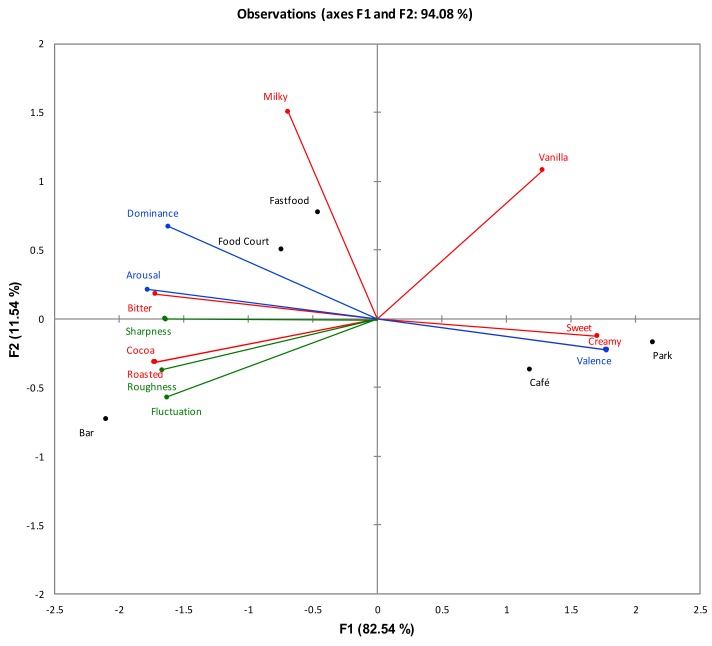
Multiple Factor Analysis (MFA) biplot depicting sensory (red vectors), affective (blue vectors) measures of chocolate gelato, and psychoacoustic (green vectors) parameters of five environmental sound conditions.

**Table 1 foods-08-00124-t001:** Sensory attributes and descriptions associated with chocolate gelati [18,20].

Attributes	Modality	Description
Sweet	Taste	Taste associated with sugar
Bitter	Taste	Taste associated with caffeine or quinine solutions
Cocoa	Flavour	Characteristic flavour associated with cocoa
Milky	Flavour	Characteristic flavour associated with milk
Creamy	Texture	Texture associated with cream
Vanilla	Flavour	A woody, slightly chemical aromatic associated with vanilla bean
Roasted	Flavour	A burnt, somewhat bitter character present in a product that has been cooked at a high temperature, typical of very strong dark coffee

**Table 2 foods-08-00124-t002:** Emotion attributes used in this study and their descriptions [40].

Emotional Reactions	Definition	Attribute Anchors
Valence	Pleasantness of the stimulus	From unpleasant to pleasant
Arousal	Intensity of emotion provoked by the stimulus	From calming to exciting
Dominance	How much does the sample grab your attention?	From controlling to not controlling your attention

**Table 3 foods-08-00124-t003:** Zwicker’s (2007) psychoacoustic parameters of the sound samples [50].

Sounds	Sharpness [acum]	Roughness [asper]	Fluctuation Strength [vacil]
Bar	4.09	0.527	2.207
Café	3.3	0.325	0.741
Fast food restaurant	3.46	0.347	0.746
Food court	3.75	0.351	1.492
Park	1.95	0.115	0.392

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
