# Peer review of "Environmental Sounds Influence the Multisensory Perception of Chocolate Gelati"

_foods, 2019, doi:10.3390/foods8040124_

Round 1
Reviewer 1 Report
It is an interesting study. There are some necessary changes to be made to improve the article.
The followings are some of the comments for the manuscript.
1. Please insert the formulation of the ice cream. Please explain why the authors have to make ice cream instead of commercially available products.
2. How many types of ice cream samples were tested?
3. How many panelists per session?
4. For ice cream 30 s break seems too short. any justification?
5. Were the sound provided simultaneously as the samples were provided or were the subjects exposed to sound first and then started tasting ice cream?
6. The experimental design is confusing. it has to be stated more clearly some where.
7. When the subjects arrived for the testing, did subjects taste one type of ice cream but exposed to 5 types of sounds? and the order of these sounds were randomized???
It is confusing when you say the serving of the sample is randomized in sample prep section.
It seems that the sound stimuli were randomized. Please be clear on the experimental design.
8. I can see that you gave 5 g of ice cream to control for TCATA. But practically serving 5 g will often end up melting. if the samples are not controlled well (especially in a plastic cup).Please describe how many sessions were held and how many subjects participated per session.
9. ln205 VAD should be written out when mentioned for the first time
10. In the data analysis.
Please delineate what a reference line is. There is a detailed description on how the reference line was determined but lacks information on what it actually is.
11. For CA analysis, please describe how sum duration of selected sensory attributes were calculated.
12. What software did the authors used for the overall data analysis?
13. Results
In the CA plots, there are all these wiggly lines. What are they for? What do they mean? What were they drawn?
14.Discussion
It will be interesting to interpret the results in terms of cross-modal congruency, (sounds-ice cream flavor) pairing familiarity, TOP appropriateness (sound quality & ice cream)
Author Response
It is an interesting study. There are some necessary changes to be made to improve the article. The followings are some of the comments for the manuscript.
1. Please insert the formulation of the ice cream. Please explain why the authors have to make ice cream instead of commercially available products.
The formulation of the ice cream can be found in Section 2.4. The chocolate gelato used in this study was specially formulated so as to slowly melt in the mouth with no strong bitter aftertaste sensations persisting before the next sample was tasted.
2. How many types of ice cream samples were tested?
Only one type of gelato was evaluated specifically a chocolate flavoured gelato formulated using the recipe in Section 2.4.
3. How many panelists per session?
All the same 58 panellists evaluated the gelato under silent and 5 different sound conditions (added to Section 2.2). We have added a sentence to aid clarity in Section 2.8. “A repeated measures design was used in this study, where all participants were exposed to all five sound conditions as well as control condition when tasting the gelato.”
4. For ice cream 30 s break seems too short. any justification?
We have added a sentence to clarify this in Section 2.4. “The time interval between samples was, in fact, determined by several pilot trials to ensure no residual carryover of flavour occurred before the next sample was tasted.”
5. Were the sound provided simultaneously as the samples were provided or were the subjects exposed to sound first and then started tasting ice cream?
Each sound was automatically played when participants first clicked the TCATA button on screen (Section 2.8). A sentence has now been added in revision to clarify this in Section 2.8. “Each sound started automatically as soon as the participants first clicked the TCATA button provided on screen.”
6. The experimental design is confusing. it has to be stated more clearly some where.
To aid clarity, we have added an overview of the experimental protocol used in this study in Section 2.8 – Figure 1.
7. When the subjects arrived for the testing, did subjects taste one type of ice cream but exposed to 5 types of sounds? and the order of these sounds were randomized??? It is confusing when you say the serving of the sample is randomized in sample prep section. It seems that the sound stimuli were randomized. Please be clear on the experimental design.
The participants tasted only one type of ice cream but consumed the ice crean under silent and five different sound conditions. The order in which they listened to the different sounds were randomised across participants. We have moved this information to Section 2.3 instead.
The sentence now reads: “Sound presentation was randomized, and counterbalanced across panellists (MacFie, Bratchell, Greenhoff, & Vallis, 1989).”
8. I can see that you gave 5 g of ice cream to control for TCATA. But practically serving 5 g will often end up melting. if the samples are not controlled well (especially in a plastic cup). Please describe how many sessions were held and how many subjects participated per session.
We have added the following sentence in Section 2.4 to clarify this. “The chocolate gelato used in this study was also specially formulated in these studies to slowly melt in the mouth with no strong bitter after taste sensations persisting before the next sample was tasted. ”
9. ln205 VAD should be written out when mentioned for the first time
VAD has been written out in full.
10. In the data analysis. Please delineate what a reference line is. There is a detailed description on how the reference line was determined but lacks information on what it actually is.
There was a mistake in Figure 2 which we have now amended, Figure 2 now contains the “bolded” highlighted line.
11. For CA analysis, please describe how sum duration of selected sensory attributes were calculated.
We have added a sentence in Section 2.9.2 according to Castura et al. (2016). “The sum duration of attributes was obtained by summing up the total CATA counts of each attribute for each product for all panellists as a function of time”
12. What software did the authors used for the overall data analysis?
We have added a sentence in Section 2.9 to clarify this: “All univariate and multivariate analysis in this study was carried out using XLSTAT (Addinsoft, U.S.A)”
13. Results. In the CA plots, there are all these wiggly lines. What are they for? What do they mean? What were they drawn?
The ‘wiggly’ lines on the CA plots are product trajectories (see Castura et al., 2016). Quoting Castura et al., 2016. “If CA is conducted on the TCATA data, the product at each time slice forms a trajectory that reveals the sensory evolution of that product over time. The points along this trajectory are interpreted as any static point would be in CA...” Hence the wiggly lines on the CA provides a simplified snapshot of the oral trajectory based on the TCATA results.
14. Discussion. It will be interesting to interpret the results in terms of cross-modal congruency, (sounds-ice cream flavor) pairing familiarity, TOP appropriateness (sound quality & ice cream)
This study involves participants eating ice cream while listening to different eating environment sounds. Sound-food pairing would be more suited for music, soundtracks and more valent sounds. Results of this study would help us understand how the eating environment influences changes in flavour, which is why the discussion revolves around eating environment settings.

Reviewer 2 Report
This is an interesting and well written article. I have made a few comments which mostly aim to clarify / provide more experimental details or are suggestions which the authors may find helpful (or not).
Clarifications required:
- 58 participants remains a fairly low number, what can the authors do / say to convince us that this is enough for real trends to emerge and robust conclusions to be drawn (for example, do the same trends persist consistently when random sub-groups are analysed separately? in particular for the TCATA work).
- Line 125: it may be worth making it clear that it is the order of the soundscapes presentation which is randomized while the sample remains the same.
- Line 127-128: is 30 seconds quite enough for panellists to, not only cleanse their palate (no cracker?), but also "shake" the affective response to the last soundscape experienced? Can you provide more details about how that transition was operated and justify its duration?
- Line 133: what product was used for training purposes? In general (line 167), were there any checks performed to ensure that the panellists were able to use the scales provided (TCATA and SAM) to convey their actual perception?
- Line 139: was the warm-up session separate from the data acquisition session or was the dummy sample presented directly before the test samples? What was the dummy sample?
- Line 154: references to articles to justify the choice of attributes; 1 or 2 lines summarising the selection process may be useful.
- Instructions to panellists: it would be good to get a more thorough account of the actual instructions the panellists were provided with, for example, were they allowed to retest (take several mouthfuls) of the sample?
- Line 180: the concept of a 9 point scale does not quite fit with that of a continuous scale. Was this a category (point) or continuous scale? This also feels at odds with the statement 163 which mentions a 15 cm unstructured line scale; I am assuming that this is in relation to the original paper rather than what was used here but it may be worth stating it clearly.
- I am assuming that the experiments took place in sensory booths as Fizz was used to record the data but there are no mentions of the panellists' environment during training and testing. Can you please provide more details?
- Line 228 and Figure 1 caption: reference to "highlighted" lines is confusing as no line appears highlighted as such.
- Lines 320-321: provide the actual p-values.
Suggestions for the authors:
- The literature review around the effect of sound in multimodal perception could be expanded on slightly to provide a general overview of the outcomes / practical implications beyond noting that a significant effect has been observed before as trends may emerge.
- In the same vein, lines 60 to 67 provide examples of studies which have used TDS and TCATA but what is the overall take home message(s) from comparing these studies which is not already expressed lines 58-59?
- It may be worth clearly stating the gap in knowledge which this study aims to fill between lines 88 and 89. What is the original contribution of this work?
- Lines 376-380: a lot is made of the significant differences observed in dominance but in terms of effect size, valence and arousal appear likely to contribute more to the differences in perception. Is it worth developing the discussion a bit more around this point?
- Lines 380 to 389: does this belong in the discussion or would this section be best placed in section 2.3. Background sounds?
Author Response
This is an interesting and well written article. I have made a few comments which mostly aim to clarify / provide more experimental details or are suggestions which the authors may find helpful (or not). Clarifications required:
58 participants remains a fairly low number, what can the authors do / say to convince us that this is enough for real trends to emerge and robust conclusions to be drawn (for example, do the same trends persist consistently when random sub-groups are analysed separately? in particular for the TCATA work).
Similarly to our previous published research, we now state that the use of 40 participants can be expected to achieve a statistical power of 0.9 - 0.95 based on Cohen’s d calculation of 0.8. The panellists are also all trained and therefore this experiment is robust. In addition, a repeated measure design was also used for this study where all 58 participants consumed ice cream under the silent and 5 sound conditions. We are confident that the number of trained panellists used in this study and the fact that a repeated measures design experiment was used makes it sufficient for robust conclusions to be drawn. In sensory descriptive analysis, a minimum of 10 trained panellists are normally required.
A sentence has been added in Section 2.2 that reads: “A minimum of 40 participants were required for each condition to achieve statistical power of 0.9 – 0.95. The calculation was based on Cohen’s d calculation of 0.8.” We have also reiterated that the 58 panellists are trained.
Line 125: it may be worth making it clear that it is the order of the soundscapes presentation which is randomized while the sample remains the same.
We have clarified this in the last 2 sentences in Section 2.
Line 127-128: is 30 seconds quite enough for panellists to, not only cleanse their palate (no cracker?), but also "shake" the affective response to the last soundscape experienced? Can you provide more details about how that transition was operated and justify its duration?
We have added a sentence in Section 2.4 to clarify this. The time interval between samples was, in fact, determined by several pilot trials to ensure no residual carryover of flavour occurred before the next sample was tasted. As soundscape was scaled to 70 dB SPL, which is a comfortable sound level, the affective response to the last soundscape experienced should not influence the following soundscape.
Line 133: what product was used for training purposes? In general (line 167), were there any checks performed to ensure that the panellists were able to use the scales provided (TCATA and SAM) to convey their actual perception?
A commercially available chocolate ice cream was used for training purposes. There were no direct checks (i.e. statistical procedure) to validate that the actual perception. However we ensured that all participants understood the meaning of each attributes (both sensory and affect) during the training and warm up sessions. A sentence has been added to Section 2.5. “A commercially available chocolate ice cream was used for training purposes”
Line 139: was the warm-up session separate from the data acquisition session or was the dummy sample presented directly before the test samples? What was the dummy sample?
A warm-up session (so called dummy session) was provided prior to the real evaluation using the real sample. The dummy sample was the same sample used in panel training. This was done to further train the participants to carry out TCATA evaluations prior to the real evaluation.
Line 154: references to articles to justify the choice of attributes; 1 or 2 lines summarising the selection process may be useful.
This has been clarified by addition of the following sentence in revision: “The attributes were selected based on a focus group who identified the most important attributes in chocolate gelati sample that changed over time.”
Instructions to panellists: it would be good to get a more thorough account of the actual instructions the panellists were provided with, for example, were they allowed to retest (take several mouthfuls) of the sample?
The instructions that were used in this study are now explained in Section 2.8. The participants were carefully instructed on-screen when the sample should be introduced into the mouth, how long it should be kept in the mouth before swallowing, and when exactly to swallow samples. They were not allowed to retaste samples.
Line 180: the concept of a 9 point scale does not quite fit with that of a continuous scale. Was this a category (point) or continuous scale? This also feels at odds with the statement 163 which mentions a 15 cm unstructured line scale; I am assuming that this is in relation to the original paper rather than what was used here but it may be worth stating it clearly.
A 9-point categorical scale was used in this study. This has been amended in the manuscript (Sections 2.7 & 2.8)
I am assuming that the experiments took place in sensory booths as Fizz was used to record the data but there are no mentions of the panellists' environment during training and testing. Can you please provide more details?
Yes, both training and experiments were done in a sensory evaluation laboratory at the university.
A sentence has been added in Section 2.2. “Both training and evaluation was done in a sensory laboratory at Auckland University of Technology”
Line 228 and Figure 1 caption: reference to "highlighted" lines is confusing as no line appears highlighted as such.
We have added the correct figure to the manuscript.
Lines 320-321: provide the actual p-values.
We have ran additional analysis on R to retrieve the actual p values, the analysis showed that the results were highly significant and the exact p value was 2.2e-16 for all affective dimensions. We believe that this might not be useful to include the exact value in the manuscript but we had amended the level of significance to < 0.001 instead to illustrate significance at 0.01%.
Suggestions for the authors: The literature review around the effect of sound in multimodal perception could be expanded on slightly to provide a general overview of the outcomes / practical implications beyond noting that a significant effect has been observed before as trends may emerge.
Changes have been made to the first paragraph of the introduction in order to provide the implications of crossmodal correpondences between sound and food perception.
In the same vein, lines 60 to 67 provide examples of studies which have used TDS and TCATA but what is the overall take home message(s) from comparing these studies which is not already expressed lines 58-59?
We agree with the reviewer that lines 60-67 only provide examples. We have deleted these lines as the take home message has already been implied in lines 58-59. We have rewritten the introduction by making changes to paragraph 2 and the final paragraph.
It may be worth clearly stating the gap in knowledge which this study aims to fill between lines 88 and 89. What is the original contribution of this work?
This was also raised by the Editor, we have made changes to the second and final paragraph in the Introduction section.
Lines 376-380: a lot is made of the significant differences observed in dominance but in terms of effect size, valence and arousal appear likely to contribute more to the differences in perception. Is it worth developing the discussion a bit more around this point?
We agree that the effects of valence and arousal are more significant compared to the dominance dimension in this study. Previous studies have confirmed the importance of valence and arousal and this is already explained in detail in paragraphs 1 and 2 of Section 4.1 of this revised manuscript. However, with eating environment sounds, dominance was also an important measure. We reaffirmed this in Section 4.4 of the discussion. As mentioned in Section 4.4., this study is one of the few studies in the crossmodal sensory science field that measured dominance in addition to valence and arousal found significant differences in dominance ratings of the different eating soundscapes used that might have influenced changes in ice cream flavour.
Lines 380 to 389: does this belong in the discussion or would this section be best placed in section 2.3. Background sounds?
This was added to the discussion section as sound level was shown to influence dominance ratings. We have made changes to the manuscript to illustrate the relationship between the two better.
